# Immunity Awareness—Strategies to Improve the Degree of Acceptance of Vaccines: A Systematic Review

**DOI:** 10.3390/vaccines13060618

**Published:** 2025-06-07

**Authors:** Alejandro Martínez-Serrano, Montserrat Pulido-Fuentes, Blanca Notario-Pacheco, Ana María Palmar-Santos, Ana Isabel Cobo-Cuenca, Ana Díez-Fernández

**Affiliations:** 1Social and Health Care Research Center, Universidad de Castilla-La Mancha, 16071 Cuenca, Spain; alejandromserrano@hotmail.com (A.M.-S.); blanca.notario@uclm.es (B.N.-P.); ana.diez@uclm.es (A.D.-F.); 2Servicio de Salud de Castilla-La Mancha, 02008 Albacete, Spain; 3Facultad de Ciencias de la Salud, Universidad de Castilla-La Mancha, 45600 Talavera de la Reina, Spain; 4Department of Nursing, Physiotherapy and Occupational Therapy, Universidad de Castilla-La Mancha, 16002 Cuenca, Spain; anaisabel.cobo@uclm.es; 5Facultad de Enfermería de Cuenca, Universidad de Castilla-La Mancha, 16002 Cuenca, Spain; 6Department of Nursing, Universidad Autónoma de Madrid, 28029 Madrid, Spain; ana.palmar@uam.es; 7Group IMCU, Faculty of Physiotherapy and Nursing, Universidad de Castilla-La Mancha, 45071 Toledo, Spain

**Keywords:** vaccination hesitancy, vaccine hesitancy, vaccines, vaccination, vaccination refusal, intervention

## Abstract

**Background/Objectives:** Vaccine hesitancy is one of the top ten threats to global health. It is necessary to develop appropriate strategies to address vaccine hesitancy. This systematic review aimed to analyze strategies used to improve the acceptance of vaccines, address doubts, and/or increase confidence and motivation in routine vaccination across all age groups. **Methods:** A systematic review was conducted of the MEDLINE, Dialnet, Scielo, CINAHL, and CENTRAL databases between 2018 and 2023. The inclusion criterion was full-text studies in English or Spanish that improve the degree of acceptance of vaccines and were evaluated by vaccination rate or pre- or postintervention tests. For data extraction, each study was categorized as community education, tailored messages, media, and new technologies. **Results:** A total of 1938 studies were identified, 38 of which were selected. New technology-based interventions used in the adult population for several vaccines offer broad reach, user interaction, and data accessibility. Tailored message strategies were used mainly among parents to foster strong relationships through respectful and empathetic dialog. Community education programs were targeted mainly at adolescents, emphasizing the use of structured, appropriate and interactive materials. Media campaigns were used as a support strategy for community education and new technology strategies due to their simplicity, wide coverage, and reach. **Conclusions:** The best strategies for reducing hesitancy are multicomponent interventions with structured and organized educational content based on the reasons for hesitancy and tailored to the target population. Therefore, caution must be taken when applying interventions, given that no single strategy can address this issue.

## 1. Introduction

In 2019, the World Health Organization (WHO) established a list of the ten main threats to global health, identifying vaccine hesitancy among them [1]. This concept has been redefined as a motivational state of being conflicted about, or opposed to, getting vaccinated [2]. Vaccine hesitancy is not homogeneous; it includes a spectrum of attitudes or sentiments that determines vaccine coverage in a lasting manner [3].

Vaccination has been one of the greatest achievements in public health, improving quality of life and reducing morbidity and mortality [4,5]. However, access and structural barriers that entrench inequalities persist [6,7], and prevent millions of people from benefiting from vaccination [5]. These inequalities also affect the level of uptake of vaccination, which has prompted global efforts to be increased [5]. It is therefore necessary to study the reasons that determine the acceptance or non-acceptance of vaccines. Whist economic and structural causes, as well as information and access to vaccines determine vaccine uptake in disadvantaged settings [6,7], social and behavioural factors need to be investigated as an equity approach.

The behavioral and social drivers (BeSD) model of vaccination are defined as beliefs and experiences specific to vaccination that are potentially modifiable to increase vaccine uptake [2]. This model has four domains and the term vaccine hesitancy is part of the Motivation domain [2,8]. In addition to the Motivation domain, we can find the domains thinking and feeling (cognitive and emotional responses, including, perceived disease risk and vaccine confidence), social processes (social norms), and practical issues (barriers, access, costs) [2,8].

In the last year, increased vaccine hesitancy has led to a decrease in the vaccination rate with an increase in the number of cases of measles in Europe [9], diphtheria in Africa [10], whooping cough in Spain [11], or the increase in anti-vaccine movements in the United States [12,13]. Among those, there is insufficient information and concern about safety [14], the presence of significant gaps and logistical barriers that reduce access to vaccination in low- and middle-income countries and regions [6,7], and the need for communication campaigns and behavioral interventions [12].

Currently, there is a considerable disparity of strategies and systematic reviews, with widely varying interventions depending on the vaccine and the target population groups [15,16]. To address this issue, the WHO Strategic Advisory Group of Experts on Immunization (SAGE) working group [16] developed an analysis which concluded that the most effective interventions were those that use multiple strategies (multi-component interventions) to enhance vaccine uptake, knowledge, and awareness.

In addition, the WHO has made a classification according to the BeSD model [2,17]: For motivation and thinking and feeling domains, to develop dialogue-based interventions and campaigns to inform or educate the public are recommended; for the social processes domain, activities such as positive social messages or recommendations to vaccinate from health workers; and for practical issues domain, is better to implement strategies like reminder for next dose, on-site vaccination at work or reminders [2,17]. Thus, following the SAGE statement, within the BeSD model, the most effective interventions are those that fall under the motivation and thoughts and feelings domains [2,16,17].

Therefore, this systematic review aimed to describe and analyze the different strategies used to improve the degree of acceptance of vaccines, address doubts about vaccination, and/or increase confidence and motivation in routine vaccination across all age groups. Additionally, it aimed to identify the primary population groups targeted by these interventions and to classify the types of interventions employed.

## 2. Materials and Methods

This study is reported according to the criteria described in the Preferred Reporting Items for Systematic Reviews and Meta-Analyses (PRISMA) 2020 declaration [18] and follows the recommendations of the Cochrane Collaboration Handbook [19]. The checklist of PRISMA reporting guidelines has also been added (Appendix A). We have registered and published this review on PROSPERO (registration number: CRD42022372461).

### 2.1. Eligibility Criteria

The study characteristics are based on the Population/Intervention/Comparator/Outcome (PICO) approach [20]. The population [P] addressed was adolescents, university students, parents/caregivers of young children, and adults in general. Interventions [I] included strategies to improve the degree of acceptance of vaccines, addressed doubts about vaccination and/or increased confidence and motivation in routine vaccination. The different strategies used control groups or vaccination rates from previous years as comparators [C]. The outcome [O] was the vaccination rate or a pre/postintervention test.

In the search for vaccination strategies, globally recommended vaccines for all age groups (children, adolescents and adults) were included, such as diphtheria, tetanus, whooping cough, hepatitis B, seasonal influenza, poliomyelitis, Hemophilus influenza type B (Hib), meningococcal, pneumococcal, tuberculosis or bacille Calmette–Guerin (BCG), measles, mumps and rubella (MMR), varicella, human papillomavirus (HPV), oral polio, and yellow fever. Articles conducted in high-income countries (HICs) and lower middle-income countries (LMICs) were included.

The main criteria were quasiexperimental design studies and randomized controlled trials published in the last 5 years (2018 to 2023), with no limit on the country, and full-text in the English or Spanish language. The languages of the articles to be analyzed were included in relation to the research team’s knowledge of them. Also, included articles to improve the degree of acceptance of vaccines, address doubts and hesitancy and/or increase confidence and motivation by assessing the effectiveness of the intervention through vaccination rates or pre/post testing.

Bibliographic reviews or opinion-based articles were excluded, as well as intervention protocols; articles that develop interventions or communication strategies aimed at improving training of healthcare workers (HCWs) to care for patients with vaccine hesitancy; articles analyzing effective education methods in the population without establishing interventions; cross-sectional studies whose objective is to know factors related to rejection or delay vaccination; articles with interventions that do not present methods to measure the level of effectiveness and Coronavirus disease 2019 (COVID-19) vaccine interventions.

### 2.2. Search Strategy and Selection Process

For a comprehensive search, the keywords “vaccination”, “vaccine”, “vaccine hesitancy”, “education”, “strategies”, and “refusal” were used in combination with the “AND” operator. The literature search was carried out in the MEDLINE, Dialnet, Scielo, CINAHL, and CENTRAL databases. From the keywords, a search string was developed for each database (Appendix A). The initial search was conducted on 24-02-2022 with subsequent updates on 20-09-2022 and 07-08-2023.

The article selection process was conducted following the PRISMA approach (Figure 1). Two investigators (A.M.-S. and M.P.-F.) reviewed the eligibility of the identified studies using the inclusion and exclusion criteria and examined the title and abstract for subsequent retrieval and analysis of the full text. Disagreements were resolved by consensus or with the involvement of a third investigator (A.D.-F.). The reasons for excluding these articles are detailed in the Appendix A.

### 2.3. Data Extraction

For data extraction and synthesis, we tabulated the data where the following characteristics of the study were included: (A) authors, year of publication, (B) study design, (C) sample size, study population and country, (D) vaccine considered, (E) information about the intervention being studied, such as the type of intervention or its content, (F) method of evaluation of efficacy and (G) study conclusions.

### 2.4. Synthesis of Results

For the analysis of the results, the data were entered into a spreadsheet and the results were divided, according to the type of intervention, into four subcategories based on the classification in the article by Singh et al. [15] using participants, interventions, comparisons, outcomes, and study design (PICO) strategy [20]:Community education: Dissemination of information adapted to specific groups of the population through health workers, mobilizers, and community defenders, proposing an education program with different sessions and activities.Tailored message: Establish a personalized and individual message, through a private visit or online.Media: Mobilization of information through different campaigns and platforms, such as radio, television, and print media.New technologies: Use of the Internet as a means of disseminating information, sharing stories and experiences and promoting the search for truthful information on the Internet.

The subcategories of the classification are not mutually exclusive; thus, the same article may contain more than one type of intervention.

### 2.5. Quality Assessment and Risk of Bias

Given the wide variability in the types of studies, three different tools were used to assess the risk of bias: (I) NIH quality assessment tool for before–after (pre–post) studies without a control group [21]; (II) the Cochrane risk-of-bias tool for randomized trials (RoB 2) [22]; and (III) the risk of bias in nonrandomized studies of interventions (ROBINS-I) tool [23]. Two reviewers (A.M.-S. and M.P.-F.) independently assessed the risk of bias in the included studies. A third reviewer (A.D.-F.) was consulted to resolve disagreements.

## 3. Results

The PRISMA diagram with the flow of studies through the review is presented in Figure 1. The search identified 1938 unreviewed articles. After deleting duplicates and those that did not meet the criteria of year of publication or language, 1362 articles were screened using the title and abstract. A total of 938 papers were excluded because the articles did not develop interventions to improve the degree of acceptance of vaccines, and 206 articles were not retrieved because the full text was not found, leaving 188 articles for full-text recovery and review. For the final review, 38 studies were use: 14 were related to community education [24,25,26,27,28,29,30,31,32,33,34,35,36,37], 8 used the tailored message method [38,39,40,41,42,43,44,45], 12 employed media [25,27,29,32,35,37,46,47,48,49,50,51], and 23 involved the intervention of new technologies [25,28,29,30,32,39,41,42,43,44,45,49,51,52,53,54,55,56,57,58,59,60,61]. Appendix A lists the main characteristics of all the studies included in the systematic review.

Regarding the date of publication, 71% of the articles selected were published in the last three years. In addition, 42.1% of the interventions involved general childhood vaccination (diphtheria, tetanus, poliomyelitis, whooping cough, Hemophilus influenzae type b (Hib), pneumo-coccus or MMR), 39.5% of which involved the HPV vaccine, and 18.5% involved the influenza vaccine. All included studies were published in English. Most of the interventions were administered to adolescents (29%) and parents of young children (37%), while only 2 articles [24,55] addressed HCWs. To assess efficacy, 37% of the studies compared vaccination rates before and after the intervention. The remaining 63% relied on pre- and post-intervention questionnaires, of which only five [29,33,47,58,60] were validated or peer-reviewed. Considering the number of types of strategies developed in each article, 22 articles involved a single type of intervention, while 16 articles involved multicomponent interventions. Table 1 shows the division of the strategies according to the number and type of intervention they received.

### 3.1. Community Education

Of the thirty-eight articles included, fourteen performed interventions based on community education activities. This type of intervention has been employed in general childhood vaccination contests, including MMR, as well as for HPV vaccination. Consequently, the majority of the analyzed studies have focused on adolescent populations [25,26,27,29,30,32,33,35], with fewer studies conducted among the parents of young children [27,28,37] and university students [31,33].

This type of intervention consisted of the development of structured health education programs designed for one or more sessions. These programs included specific content, personalized activities, and adaptations tailored to the target population [24,25,26,27,29,30,31,32,33,34,35,36,37]. These interventions were typically implemented through in-person sessions conducted primarily at schools [25,26,27,29,30,32,34], hospitals [24,36], and other community centers [28]. Additionally, online platforms such as Webex or Moodle [31] and resources such as mHealth or e-mail [28,33] were used. Commonly, written materials were provided to both the adolescents and parents of young children [25,27,29,35], and audiovisual resources, such as videos [29,30,32], were used to reinforce the knowledge acquired during these educational programs.

The implementation of community education interventions had a positive impact, resulting in an increase in population motivation to encourage their friends and family to get vaccinated [26], an increase in intentions to receive the vaccine, more widespread knowledge about vaccination, vaccination rates, and a reduction in fear [27,28,29,30,31,34,36]. Additionally, a statistically significant increase in knowledge was observed between the period before and after the adoption of the strategies [24,27,31] and between those groups that received tailored and developed interventions increase in the control arm [33]. However, some studies have reported no significant increase in vaccination rates [35].

Several articles [25,32] have employed the use of the CARD system. The comfort–ask–relax–distract (CARD) system is a vaccine delivery framework that integrates evidence-based interventions to reduce stress-related responses and improve the vaccination experience for students receiving vaccinations at school [32]. This system operates through in-class lessons and the use of distracting elements such as games or verbal support provided by classmates [25]. In addition, reminders and brochures with general information on vaccines for parents were included. It has been demonstrated that the CARD reduces stress-related responses in students receiving vaccinations at school, and these responses are positively received by students and public health staff [32].

### 3.2. Tailored Message

Eight articles [38,39,40,41,42,43,44,45] used interventions based on tailored message methods on adult populations, mainly on future parents or parents whose children were under 16 years of age. More than half of the studies that targeted parents developed interventions for general vaccines administered during childhood and adolescence [32,33,35,37]. The remaining studies have developed interventions for HPV and influenza vaccines in the adult population [40,42,44,45].

These interventions have developed through face-to-face interactions in clinic offices, hospital rooms, community locations, or home visits [38,40]. The remaining articles utilized web-based interventions, creating individually tailored web pages or videos based on information gathered from a baseline survey [39,41,42,43,44,45]. In particular, three articles have employed the MomsTalkShots application [41,42,43]. The MomsTalkShots is an individually tailored educational app that provides algorithmically personalized videos that take into account parents’ specific needs, attitudes, beliefs, and intentions related to vaccines.

When we analyzed the articles concerning the methods used to develop tailored messages, we were able to identify two main approaches: (I) utilizing structured and standardized scripts tailored to address the specific reasons for vaccine hesitancy and the characteristics of the patients [39,40,45]; and (II) employing communication strategies to design messages aimed at achieving persuasive communication and behavioral change, such as motivational interviewing (MI), the elaboration likelihood model (ELM), or the extended parallel processing model (EPPM) [38,41,42,43,44].

The findings from these studies indicate that the implementation of tailored message techniques leads to various positive outcomes. It increased the intention to get vaccinated [38,40,45], improved knowledge levels [41,42] and information-seeking behavior [44], and increased the perceived risk of vaccine-preventable diseases [43]. The lack of statistically significant differences was only found in a few studies [39,40,45].

### 3.3. Media

The use of the media as an intervention to reduce concerns about vaccination has been identified in twelve articles [25,27,29,32,35,37,46,47,48,49,50,51]. These interventions have mainly focused on the HPV vaccine [27,29,35,46,47,49,50] and, to a lesser extent, on general childhood vaccines [25,32,37,48] and influenza [51]. For this reason, these articles have primarily focused on adolescents, with less emphasis on parents and university students, such as community education articles.

In terms of the nature of the intervention, most of the articles used the printed copy (handout, flyer, leaflet, mail, newspaper article) as the main method of communication [25,27,29,32,35,37,46,47,49,50,51], and only one article used phone calls and short messaging service (SMS) [48]. Other media such as posters (posters, photography, bill-boards) or audio visual media (radio and television) were not used in the interventions analyzed.

Notably, this type of intervention was used in combination with another type of intervention in most of the articles [25,27,29,32,35,37] and with new technologies [25,29,32,49,51]; four articles that used only media intervention were found [46,47,48,50].

Some articles have reported no statistically significant differences after intervention [35,50] or have preferred alternative intervention methods over media-based approaches [37,48,51]. However, this type of strategy results in a significant positive change in beliefs regarding vaccination safety, effectiveness, knowledge, and importance in preventing illness [25,27,29,32,47,49], and it also increases vaccine series initiation and completion rates [46].

### 3.4. New Technologies

Intervention techniques based on the development of new technologies were identified in twenty-three papers analyzed in this review [25,28,29,30,32,39,41,42,43,44,45,49,51,52,53,54,55,56,57,58,59,60,61]. Given the large number of items in this category, it is necessary to perform an analysis according to the target population:Adolescents: Five articles [25,29,30,32,57] focused on school students as a target population. For this group, new technologies, such as the HPV and MMR vaccines, have been used in combination with community education and media interventions and have focused on vaccines for general childhood immunization. Videos are the primary means of message delivery and are accessible both in the classroom and externally through websites or mobile apps.Adult population: Out of the 15 articles identified for parents of young children, adults, and HCWs, all of which addressed general childhood vaccination, HPV vaccination and influenza; six of them [39,41,42,43,44,45] integrated tailored message interventions with new technologies. These interventions included the utilization of websites and mobile applications, such as MomsTalkShots, to deliver personalized information through video or messaging.In contrast, the remaining articles [28,53,54,55,56,58,59,60,61] exclusively relied on interventions focused on new technologies. These papers utilized various methods for disseminating information, including social media and applications, such as Facebook, WhatsApp, and HPVcancerFree [54,58,60]; private intranet platforms; and tools, such as hospital intranets, survey platforms or mHealth [28,53,55,59,61]; and email [56]. Information transmission methods include the use of audio visual media such as videos, interactive games (e.g., VR), and vaccine choice experiments [53,55,56,60,61]; the utilization of forums for posting queries and obtaining expert responses [54]; and the publication of written content on webpages, including advertisements, posts, articles, surveys or push messages [28,53,54,55,58,59].University students: We found three articles [49,51,52] targeting university students, that focused on HPV and influenza vaccination. They combined new technology with media-based intervention utilizing methods such as video usage and written information distribution, mainly through email and social media.

When examining the results of articles exclusively using new technology-based methods, many have reported an increase in willingness to receive vaccines [52,59,61], a greater impact when information is conveyed through immersive media platforms [53], an increase in vaccination rates [54,55], heightened knowledge and awareness regarding vaccinations, and efforts to counter vaccine misinformation [56,57,58,60].

### 3.5. Risk of Bias

Of a total of 38 studies, six were assessed with the NIH quality assessment tool for before–after (pre–post) studies without a control group, 8 were evaluated with the ROBINS-I tool, and 24 were assessed with the RoB 2 tool (Appendix A). Of the papers assessed with the ROBINS-I tool, six noted a moderate risk of bias, and two noted serious risk of bias. Whereas studies evaluated with the RoB 2 tool, fifteen reported a low risk of bias, six studies noted some concerns, and three studies reported a high risk of bias.

## 4. Discussion

This systematic review aimed to describe and analyze different strategies to improve the degree of acceptance of vaccines, address doubts about vaccination, and/or increase confidence and motivation. We found that the most used strategy was new technology-based interventions in single or multicomponent interventions. Tailored message strategies and community education programs also provided several advantages directed mainly at parents and adolescents and media campaigns were used as a support strategy for community education and new technology strategies. The search was focused on interventions that increase vaccination for the motivation and thinking and feeling domains introduced by the BeSD model; however, we identified in some types of interventions, characteristic of other domains (Table 2).

The analysis of strategies revealed that all the articles employed two distinct methods to assess the effectiveness of interventions:Vaccination coverage is employed by several reviews as an effective method of assessment, as it serves as a prime indicator of protection against vaccine-preventable diseases [62]. However, receiving a vaccine does not necessarily eliminate hesitancy, as demonstrated by the findings of Willis et al. in a survey of COVID-19-vaccinated individuals, in which 60% expressed some level of hesitancy after being vaccinated [63]. Compliance with vaccination should not be equated with the absence of hesitancy, as this overlooks ongoing concerns individuals may have before or after vaccination [63,64,65].Pre–post surveys are also used to assess interventions. However, our analysis found that most studies relied on self-developed measurement scales, which reduces homogeneity and limits comparability. Five articles [29,33,47,58,60] used validated or peer-reviewed surveys, such as the Parent Attitudes about Childhood Vaccines (PACV) [66] or HPV Adolescent Vaccine Intervention Questionnaire (HAVIQ) [67]. Interventions aimed at improving vaccine acceptance must be tailored to the specific target population, geographic context, and other relevant factors. For this reason, they are not directly comparable, as each one assesses vaccine hesitancy in different populations and for different vaccines.

Our review analyzed interventions targeting the motivation, thinking, and feeling domains of the BeSD model of vaccination. However, we found a large variability of interventions targeting the remaining domains of the BeSD model [2,15]:The use of incentives helps during the COVID-19 pandemic to accelerate vaccination [68,69,70]. However, in low- and middle-income countries, monetary incentives can exploit economic vulnerability, intensify doubts, about vaccine safety and governmental authority, and foster resentment among populations who complied with their duty to vaccinate without incentives [69]. In addition, incentives create expectations for compensation in other areas, while persuasion-based interventions are the least effective in reducing vaccine hesitancy [16,69,71].Countries like France, Italy, or Portugal have developed compulsory vaccination policies, justified by a favorable balance of risks, scientific evidence, and the reduction in inequalities [72,73]. However, such policies infringe on personal autonomy, provoke backlash, and raise exemption rates [74,75,76]. Our findings align with the Spanish bioethics committee’s recommendations, which advocate for evidence-based educational, communicative, and behavioral strategies [12,77].The modern landscape, marked by technological advances and the rise of social media, has driven the development of innovative strategies, such as the use of tools like artificial intelligence and video games, as well as the mobilization of online influencers [78,79,80].In our review, there are only two interventions aimed at addressing hesitancy among HCWs. In addition to the interventions we have mentioned, there are strategies aimed at HCWs to improve their awareness and implementation of best immunization practices. These include team discussions and communication training using a “presumptive communication approach” or motivational interviewing [81,82,83], weekly educational flyers [84], and role-playing or virtual reality simulations [85,86,87].

The present study classification method followed the division of Singh et al. [15], but the literature offers other systems. Jarret et al. [16] categorized articles into dialog-based, incentive-based, reminder–recall-based, or multicomponent interventions. Nour [88] classified interventions based on technological, mass marketing, campaigning, or direct communication strategies. MacDonald et al. [89] provided two lists with twelve evidence-informed strategies at the vaccine program and individual patient levels. Some, such as Dubé et al. [90], did not develop this classification because the number of interventions similar enough to be grouped together was low. The variety among the different types of classifications underlines the idea that a large number of interventions can be applied to reduce vaccination concerns.

However, the classification of vaccine interventions does not consider the specific characteristics of each intervention, such as the target population, type of vaccine, reasons for hesitancy, and country. Overcoming vaccine hesitancy is a complex process that necessitates identification, diagnosis, and tailored intervention that addresses cognitive, emotional, and skill-based learning areas with the goal of achieving vaccine literacy [89,91]. Vaccine literacy entails the ability to understand health information and services, empowering individuals with the skills and knowledge needed to make appropriate decisions through effective communication on vaccine efficacy, safety, and potential side effects, and individual protective benefits [92,93].

Consequently, to improve evaluation and facilitate the meaningful comparisons of short- and medium-term impacts, it is necessary to consider the domains of the BeSD model and use validated pre-post intervention surveys designed to measure vaccination hesitancy in various populations and vaccines, such as the Vaccination Confidence Index [94] or the Vaccination Attitudes Examination (VAX) [95]. Our study explores the effectiveness of different interventions that improve vaccine acceptance, addressing the causes and different dimensions behind vaccine hesitancy as an essential prior step for the formulation of public health policies to improve vaccine coverage in a context-sensitive manner [96], which improve trust in healthcare professionals’ interventions [97]. Further research is needed to evaluate the long-term objectives of interventions aimed at reducing vaccine hesitancy, considering vaccination rates in order to track trends within the general population.

Apart from the impossibility of comparing different interventions, one of the main limitations we can find in the review is the lack of literature analyzing different types of populations together with the search of articles only in English and Spanish, which could limit the validity of our results. Other reviews [15,16] have reported the presence of populations whose hesitancy was due to the influence of religious leaders, in which interventions were used in which leaders and health professionals were involved in a coordinated way. However, this reason for hesitancy was not mentioned in any of the articles reviewed and was not analyzed in the review. In addition, the different studies analyzed have been developed in different countries, implying that different cultural contexts may influence the concept of immunization; and there are other factors that may influence hesitancy that were not analyzed in the review, such as educational and socioeconomic level.

Another limitation is the inclusion of five studies assessed as having a high or serious risk of bias. Although this was considered in our analysis, these studies did not have a significant impact on the overall results. Their primary role was to support findings already established by other, more robust studies. As such, their inclusion did not alter the main outcomes or conclusions of our review.

Finally, our review focused on interventions targeting the motivation and thinking and feeling domains of the BeSD model of vaccination, as they seem to be more effective [16]. However, elements of other domains were noted in the interventions, suggesting that future research should consider these for more comprehensive results.

## 5. Conclusions

Vaccine hesitancy is a serious public health problem that puts many vulnerable people at risk. Studies investigating the factors contributing to low vaccination rates highlight the importance of understanding and addressing the underlying causes, as well as enhancing public health education to increase knowledge about vaccines and reduce hesitancy. Although most of the strategies analyzed had a positive effect, caution should be exercised in drawing general conclusions about the effectiveness of the interventions due to variations between populations, vaccines, and methods of design and evaluation.

Vaccine hesitancy is a complex issue, and no standalone strategy can fully address it. The design of an effective intervention to improve vaccination uptake must include understanding the reasons for hesitancy, developing structured and organized content, tailoring vaccination to the target population and proposing a vaccine, using validated surveys for assessment, and employing different types of interventions to achieve greater effectiveness.

These methods have been shown to enhance trust in healthcare professionals and thus their recommendations. Public health initiatives should take these factors into account to address and reduce vaccine hesitancy more effectively.

## Figures and Tables

**Figure 1 vaccines-13-00618-f001:**
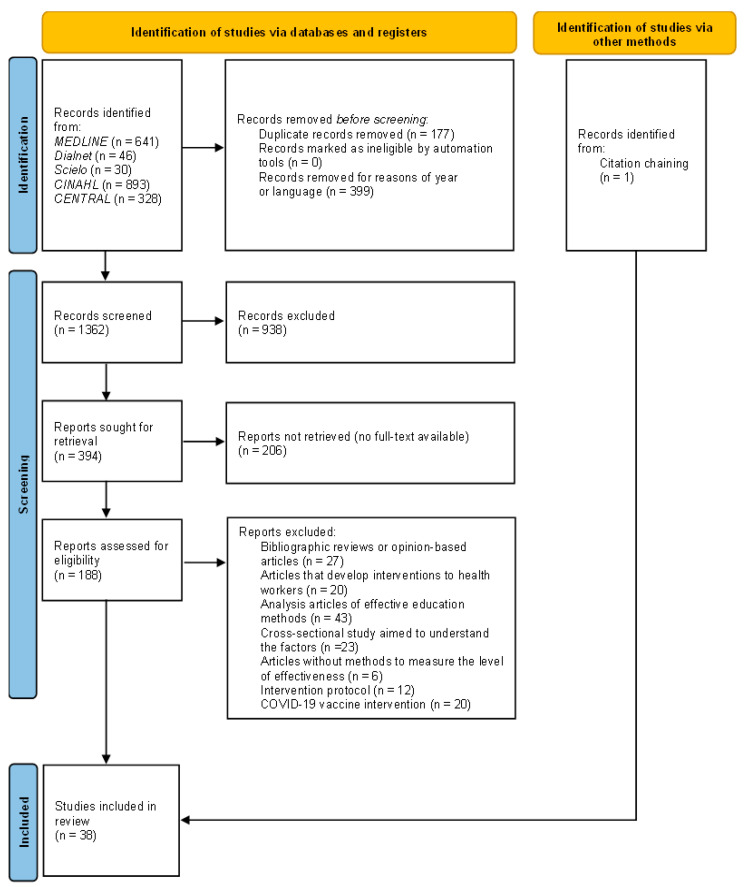
Prisma flow diagram of the complete search process.

**Table 1 vaccines-13-00618-t001:** Strategies and target population of the included studies according to the type of intervention.

Type of Intervention	Number of Articles	Target Population
Single intervention		
Community education	6	Adolescents, USs, adults, and HCWs
Tailored message	2	Adults and parents
Media	4	Adolescents, USs, and parents
New technologies	10	Adults, USs, parents, and HCWs
Multicomponent intervention		
Community education + media	3	Adolescents and parents
Community education + New technologies	2	Adolescents and parents
Tailored message + new technologies	6	Adults and parents
Media + new technologies	2	USs
Community education + media + New technologies	3	Adolescents

USs: University students, HCWs: healthcare workers.

**Table 2 vaccines-13-00618-t002:** Main population, advantages, and related BeSD domain of each type of intervention.

Type of Intervention	Main Population	Advantages	BeSD Domain
Community education	Adolescents	Structured, appropriate and interactive activities Feedback and interaction with participants	Motivation
Tailored message	Parents	Strong patient–HCW relationships Respectful and empathetic dialog Personalized messaging	Social processes
Media	-	Support strategy to other intervention Simplicity Extensive coverage Mobility reach	Motivation, social processes, thinking and feeling, practical issues
New technologies	Adults	Broad reach Affordability User interaction Data accessibility and collaboration Enriching resources	Thinking and feeling, practical issues

## Data Availability

The data presented in this study are available in this article and the Appendix A.

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
