# Peer review of "Immunity Awareness—Strategies to Improve the Degree of Acceptance of Vaccines: A Systematic Review"

_vaccines, 2025, doi:10.3390/vaccines13060618_

Round 1

Reviewer 1 Report

Comments and Suggestions for Authors

The manuscript “Immunity awareness – strategies to improve the degree of acceptance of vaccines: a systematic review” is a well-structured and scientifically written article that justifies the authors’ claims with supporting evidence. Authors adhered to currently accepted scientific standards when conducting meta-analyses, performed a comprehensive search and selection of applicable articles, systematically assessed and categorized data into four buckets, and appropriately assessed chosen articles for risk of bias. The language and terminology are precise and formal, the manuscript follows a standard structure and flow, and the in-text citations are correctly used. To enhance the impact of and clarify the conclusions made, the authors should consider the minor points listed below.

If the data included in articles analyzed were consistent across various demographics and social circumstances, the authors should consider providing more specific examples or highlighting the most effective combinations of intervention types. The nuanced findings presented within the review could more explicitly articulate the direct implications for public health policy-makers, practitioners, or individuals communicating the benefits of vaccination.

The manuscript summarizes a great deal of extensive research looking into vaccine hesitancy. Some of the important findings within the manuscript may get lost among the text. The authors should consider incorporating a more quantitative statement and visual representation of the findings from the assessed datasets.

The authors make a very important point between lines 343 and 348. Vaccine hesitancy and vaccine uptake are two distinct metrics that can be impacted by a multitude of factors and do not necessarily share a linear relationship. In an ideal world, a relative decrease in vaccine hesitancy would directly correlate with an increase in vaccine uptake. Considering the articles reviewed, were any vaccine interventions (alone or in combination) shown to successfully translate a decrease in hesitancy into a direct increase in vaccine uptake?

Comments on the Quality of English Language

Perhaps a formatting issue, but words are hyphenated multiple times throughout the manuscript. A few examples: Line 50 (vac-cines), Line 57 (do-mains), Line 276 (im-portance).

Author Response

Point-by-point response to Comments and Suggestions for Authors

Comments 1: The manuscript “Immunity awareness – strategies to improve the degree of acceptance of vaccines: a systematic review” is a well-structured and scientifically written article that justifies the authors’ claims with supporting evidence. Authors adhered to currently accepted scientific standards when conducting meta-analyses, performed a comprehensive search and selection of applicable articles, systematically assessed and categorized data into four buckets, and appropriately assessed chosen articles for risk of bias. The language and terminology are precise and formal, the manuscript follows a standard structure and flow, and the in-text citations are correctly used. To enhance the impact of and clarify the conclusions made, the authors should consider the minor points listed below.

If the data included in articles analyzed were consistent across various demographics and social circumstances, the authors should consider providing more specific examples or highlighting the most effective combinations of intervention types. The nuanced findings presented within the review could more explicitly articulate the direct implications for public health policy-makers, practitioners, or individuals communicating the benefits of vaccination.

Response 1: Thank you for pointing this out. We have faced several circumstances due to the variability of the populations included, type of interventions, etc, thus, to state clear conclusions about the best intervention should be taken with caution.

Trying to make this clearer, we have modified the discussion and conclusion section to articulate the direct implications of our results and to enhance the impact of the manuscript. It can be found in page 11, lines 405-409, 435-438, 444, 447-449:

“Our study explores the effectiveness of different interventions that improve vaccine acceptance, addressing the causes and different dimensions behind vaccine hesitancy, as an essential prior step for the formulation of public health policies to improve vaccine coverage in a context-sensitive manner [97], which improve trust in healthcare professionals' interventions [98].”.

“Studies investigating the factors contributing to low vaccination rates highlight the importance of understanding and addressing the underlying causes, as well as enhancing public health education to increase knowledge about vaccines and reduce hesitancy.”.

“using validated surveys for assessment”

“These methods have been shown to enhance trust in healthcare professionals and their recommendations. Public health initiatives should take these factors into account to more effectively address and reduce vaccine hesitancy.”.

Comments 2: The manuscript summarizes a great deal of extensive research looking into vaccine hesitancy. Some of the important findings within the manuscript may get lost among the text. The authors should consider incorporating a more quantitative statement and visual representation of the findings from the assessed datasets.

Response 2: Agree. We have, accordingly, included a table with the main advantages of each type of interventions according to our results to emphasize this point and summarise them. This change can be found in pages 8 and 9, lines 325-332 and line 334, Table 2.

Comments 3: The authors make a very important point between lines 343 and 348. Vaccine hesitancy and vaccine uptake are two distinct metrics that can be impacted by a multitude of factors and do not necessarily share a linear relationship. In an ideal world, a relative decrease in vaccine hesitancy would directly correlate with an increase in vaccine uptake. Considering the articles reviewed, were any vaccine interventions (alone or in combination) shown to successfully translate a decrease in hesitancy into a direct increase in vaccine uptake?

Response 3: We agree with the commentary. The aim of all the articles analysed is to reduce vaccine hesitancy by developing dialogue-based interventions and campaigns to inform and educate the public. As an effectiveness metric, some articles have made a comparison of the vaccination rate, with a significant increase observed in several instances (Di Mauro, A et al. 2021 [54] or Lecce, M et al. 2021 [55], both being strategies based on new technologies). However, as pointed out on page 9, lines 337-344, vaccination rates are more appropriate for evaluating long-term objectives, but the majority of analysed articles use very short analysis periods. It would be interesting in future work to carry out the analysis to see which intervention is more effective over a longer evaluation period.

Additionally, we have included a paragraph in the Discussion section acknowledging the need to design studies that enable meaningful comparisons. Please see page 9, lines 335-353:

“The analysis of strategies revealed that all the articles employed two distinct methods to assess the effectiveness of interventions:

  • Vaccination coverage is employed by several reviews as an effective method of assessment, as it serves as a prime indicator of protection against vaccine-preventable diseases [62]. However, receiving a vaccine does not necessarily eliminate hesitancy, as demonstrated by the findings of Willis et al. in a survey of COVID-19-vaccinated individuals, in which 60% expressed some level of hesitancy after being vaccinated [63]. Compliance with vaccination should not be equated with the absence of hesitancy, as this overlooks ongoing concerns individuals may have before or after vaccination [63–65].
  • Pre-post surveys are also used to assess interventions. However, our analysis found that most studies relied on self-developed measurement scales, which reduces homogeneity and limits comparability. Only five articles [29,33,47,58,60] used validated or peer-reviewed surveys, such as Parent Attitudes about Childhood Vaccines (PACV) [66] or HPV Adolescent Vaccine Intervention Questionnaire (HAVIQ) [67]. Interventions aimed at improving vaccine acceptance must be tailored to the specific target population, geographic context, and other relevant factors. For this reason, they are not directly comparable, as each one assesses vaccine hesitancy in different populations and for different vaccines.”.

And please see pages 10 and 11, lines 401-412:

“Consequently, to improve evaluation and facilitate meaningful comparisons of short- and medium-term impacts, it is necessary to consider the domains of the BeSD model and use validated pre-post intervention surveys designed to measure vaccination hesitancy in various populations and vaccines, such as the Vaccination Confidence Index [95] or the Vaccination Attitudes Examination (VAX) [96]. Our study explores the effectiveness of different interventions that improve vaccine acceptance, addressing the causes and different dimensions behind vaccine hesitancy, as an essential prior step for the formulation of public health policies to improve vaccine coverage in a context-sensitive manner [97], which improve trust in healthcare professionals' interventions [98]. Further research is needed to evaluate the long-term objectives of interventions aimed at reducing vaccine hesitancy, considering vaccination rates in order to track trends within the general population.”.

Response to Comments on the Quality of English Language

Point 1: Perhaps a formatting issue, but words are hyphenated multiple times throughout the manuscript. A few examples: Line 50 (vac-cines), Line 57 (do-mains), Line 276 (im-portance).

Response 1: We are sorry for the mistakes. We believe it was a formatting issue, and we have corrected them. Thank you for noticing and we are sorry for the inconvenience.

Reviewer 2 Report

Comments and Suggestions for Authors

This systematic review evaluates strategies to improve vaccine acceptance, addressing hesitancy across diverse populations. Analyzing 38 studies (2018–2023), the authors highlight multicomponent interventions (e.g., combining community education, tailored messaging, and technology) as most effective. Key findings include: community education enhances adolescent engagement through structured programs; tailored messages improve parental trust via empathetic communication; and technology-based tools (e.g., apps, videos) offer broad reach and interactivity. Media campaigns showed limited standalone efficacy but synergized with other strategies. The review underscores the need for context-specific, population-tailored interventions and identifies gaps in addressing cultural, socioeconomic, and structural barriers. Its significance lies in synthesizing actionable insights for policymakers and healthcare providers to combat global vaccine hesitancy.

Major Issues

  1. Population Representativeness: The focus on high-income countries (HICs) and English/Spanish studies limits generalizability. Include LMICs and non-Western contexts to address global vaccine equity gaps.
  2. Risk of Bias: High/Serious bias in 5/38 studies (per ROBINS-I/RoB 2 tools) is inadequately addressed, potentially skewing conclusions.
  3. BeSD Model Integration: While citing the BeSD framework, the analysis minimally explores interventions targeting the Social Processes and Practical Issues domains (e.g., policy, access barriers). Expand this discussion.
  4. Long-Term Outcomes: Most studies measure short-term metrics (e.g., knowledge). Assess long-term impacts on vaccination rates and sustained behavior change.
  5. Intervention Comparability: Heterogeneity in interventions (e.g., vaccine types, delivery methods) hinders direct comparisons. Use standardized outcome measures (e.g., the Vaccine Confidence Index).
  6. Ethical Considerations: Incentive-based strategies (briefly mentioned) raise equity concerns. Discuss ethical implications of monetary incentives in LMICs.

Minor Issues

  1. COVID-19 Exclusion: The rationale for excluding COVID-19 vaccine studies is unclear, given its relevance to hesitancy research. Justify this choice or update the scope.
  2. HCW Engagement: Only 2 studies involve healthcare workers (HCWs), a critical group for vaccine promotion. Recommend HCW training interventions as a priority.
  3. Cultural Nuances: Neglects religious/ethnic drivers of hesitancy (e.g., role of community leaders). Incorporate case studies from diverse cultural contexts.
  4. Technology Access: New technology interventions assume digital literacy. Address accessibility barriers (e.g., rural areas, low-income groups).
  5. Publication Bias: Over 70% of studies are from the past 3 years. Ensure older studies are not disproportionately excluded if methodologically robust.
  6. Visual Aids: Expand tables (e.g., Table 1) to include intervention efficacy metrics (e.g., vaccination rate changes) for clearer comparisons.

Author Response

Point-by-point response to Comments and Suggestions for Authors

Comments 1: This systematic review evaluates strategies to improve vaccine acceptance, addressing hesitancy across diverse populations. Analyzing 38 studies (2018–2023), the authors highlight multicomponent interventions (e.g., combining community education, tailored messaging, and technology) as most effective. Key findings include: community education enhances adolescent engagement through structured programs; tailored messages improve parental trust via empathetic communication; and technology-based tools (e.g., apps, videos) offer broad reach and interactivity. Media campaigns showed limited standalone efficacy but synergized with other strategies. The review underscores the need for context-specific, population-tailored interventions and identifies gaps in addressing cultural, socioeconomic, and structural barriers. Its significance lies in synthesizing actionable insights for policymakers and healthcare providers to combat global vaccine hesitancy.

Major Issues

  1. Population Representativeness: The focus on high-income countries (HICs) and English/Spanish studies limits generalizability. Include LMICs and non-Western contexts to address global vaccine equity gaps.
  2. Risk of Bias: High/Serious bias in 5/38 studies (per ROBINS-I/RoB 2 tools) is inadequately addressed, potentially skewing conclusions.
  3. BeSD Model Integration: While citing the BeSD framework, the analysis minimally explores interventions targeting the Social Processes and Practical Issues domains (e.g., policy, access barriers). Expand this discussion.
  4. Long-Term Outcomes: Most studies measure short-term metrics (e.g., knowledge). Assess long-term impacts on vaccination rates and sustained behavior change.
  5. Intervention Comparability: Heterogeneity in interventions (e.g., vaccine types, delivery methods) hinders direct comparisons. Use standardized outcome measures (e.g., the Vaccine Confidence Index).
  6. Ethical Considerations: Incentive-based strategies (briefly mentioned) raise equity concerns. Discuss ethical implications of monetary incentives in LMICs.

Response 1: We appreciate the reviewer concern regarding these issues. We have, accordingly, revised them, as follows:

  1. Population Representativeness: Thank you for this comment. In the inclusion criteria, both HICs and LMICs were included and the search retrieved results from a broad range of non-occidental countries like Iran, India, China, Australia, Lebanon, Japan, Indonesia and Nigeria. But we agree with the reviewer that is difficult to ensure a full representativeness, even more taking into account the difficulties to conduct research and to publish them in certain population and countries. We have addressed this issue in the limitations, Please see pages 11, lines 413-415;420-423:

“…one of the main limitations we can find in the review is the lack of literature analyzing different types of populations together with the search of articles only in English and Spanish, which could limit the validity of our results”… “In addition, the different studies analyzed have been developed in different countries, implying that different cultural contexts may influence the concept of immunization; and there are other factors that may influence hesitancy that were not analyzed in the review, such as educational and socioeconomic level.”.

  1. Risk of Bias: We agree that this implies a limitation of our results. We found out that one of the main problems was the randomization process and we took it into account when analyzing the results and with the possible implications. However, during the analysis, articles with high or severe bias did not introduce new ideas into the manuscript. Instead, they supported arguments established by other interventions. We have addressed this issue in the limitations, Please see page 11, lines 424-428:

“Another limitation is the inclusion of five studies assessed as having a high or serious risk of bias. Although this was considered in our analysis, these studies did not have a significant impact on the overall results. Their primary role was to support findings already established by other, more robust studies. As such, their inclusion did not alter the main outcomes or conclusions of our review.”.

  1. BeSD Model Integration: We appreciate this comment since is something we have discussed among the authors as well. We decided not to include the Social Processes and Practical Issues, because the articles discussing these domains of the BeSD model are not considered among the most effective by the WHO Strategic Advisory Group of Experts on Immunization (SAGE) working group, as stated in the introduction section, Please see page 2, lines 70-73; 79-81:

“… the WHO Strategic Advisory Group of Experts on Immunization (SAGE) working group [16] developed an analysis which concluded that the most effective interventions were those that use multiple strategies (multicomponent interventions) to enhance vaccine uptake, knowledge, and awareness.“…“Thus, following the SAGE statement, within the BeSD model, the most effective interventions are those that fall under the Motivation and Thoughts and feelings domains [2,16,17].”.

Besides, we consider these domains is not applicable with the aim of this systematic review, which is to describe and analyze the different strategies used to improve the degree of acceptance of vaccines, address doubts about vaccination and/or increase confidence and motivation in routine vaccination across all age groups. 

However, these domains could not be left aside, since, in fact, different interventions (new technologies, tailored message, etc.) can be included in this domain and thus, is mentioned in the text, Please see page 9; Table 2.

The evolution of BeSD domains in the developing of new vaccination intervention will allow us to make, in the future, a full comprehensive picture of this behavioral and social drivers of vaccination model.

  1. Long-Term Outcomes and 5. Intervention Comparability: Thank you for pointing this out. One of the main difficulties in analyzing these articles was the methodological heterogeneity in the assessment of efficacy. Some articles used pre- and post-intervention surveys for short- and medium-term analysis, of which most used either self-developed or very specific validated surveys, making inter-article comparisons difficult. The alternative method was to compare vaccination rates (Di Mauro, A et al. 2021 [54] or Lecce, M et al. 2021 [55], both being strategies based on new technologies) revealing significant differences only in long-term settings. This heterogeneity complicates comparisons between articles. Therefore, we recommend using validated surveys applicable to general populations and vaccines and using vaccination rates for long-term comparative analyses.

We have included information about the evaluation method used by the articles in the results, Please see page 6, lines 190-193:

“To assess efficacy, 37% of the studies compared vaccination rates before and after the intervention. The remaining 63% relied on pre- and post-intervention questionnaires, of which only five [29,33,47,58,60] had been validated or peer-reviewed.”.

Also, in the discussion section addresses this issue and we include a paragraph on this subject, Please see page 9, lines 335-353:

“The analysis of strategies revealed that all the articles employed two distinct methods to assess the effectiveness of interventions:

    • Vaccination coverage is employed by several reviews as an effective method of assessment, as it serves as a prime indicator of protection against vaccine-preventable diseases [62]. However, receiving a vaccine does not necessarily eliminate hesitancy, as demonstrated by the findings of Willis et al. in a survey of COVID-19-vaccinated individuals, in which 60% expressed some level of hesitancy after being vaccinated [63]. Compliance with vaccination should not be equated with the absence of hesitancy, as this overlooks ongoing concerns individuals may have before or after vaccination [63–65].
    • Pre-post surveys are also used to assess interventions. However, our analysis found that most studies relied on self-developed measurement scales, which reduces homogeneity and limits comparability. Only five articles [29,33,47,58,60] used validated or peer-reviewed surveys, such as Parent Attitudes about Childhood Vaccines (PACV) [66] or HPV Adolescent Vaccine Intervention Questionnaire (HAVIQ) [67]. Interventions aimed at improving vaccine acceptance must be tailored to the specific target population, geographic context, and other relevant factors. For this reason, they are not directly comparable, as each one assesses vaccine hesitancy in different populations and for different vaccines.

And please see pages 10 and 11, lines 401-412:

“Consequently, to improve evaluation and facilitate meaningful comparisons of short- and medium-term impacts, it is necessary to consider the domains of the BeSD model and use validated pre-post intervention surveys designed to measure vaccination hesitancy in various populations and vaccines, such as the Vaccination Confidence Index [95] or the Vaccination Attitudes Examination (VAX) [96]. Our study explores the effectiveness of different interventions that improve vaccine acceptance, addressing the causes and different dimensions behind vaccine hesitancy, as an essential prior step for the formulation of public health policies to improve vaccine coverage in a context-sensitive manner [97], which improve trust in healthcare professionals' interventions [98]. Further research is needed to evaluate the long-term objectives of interventions aimed at reducing vaccine hesitancy, considering vaccination rates in order to track trends within the general population.”.

  1. Ethical Considerations: We agree with that affirmation. We initially included references to both low- and middle-income countries as well as high-income countries, however, we have modified the paragraph in the discussion where this issue is more clearly addressed, Please see page 10, lines 359-364:

“However, in low- and middle-income countries monetary incentives can exploit economic vulnerability, intensify doubts about vaccine safety and governmental authority, and foster resentment among populations who complied with their duty to vaccinate without incentives [69]. In addition, incentives create expectations for compensation in other areas, while persuasion-based interventions are the least effective in reducing vaccine hesitancy [69,71,72].”.

Comments 2: Minor Issues

  1. COVID-19 Exclusion: The rationale for excluding COVID-19 vaccine studies is unclear, given its relevance to hesitancy research. Justify this choice or update the scope.
  2. HCW Engagement: Only 2 studies involve healthcare workers (HCWs), a critical group for vaccine promotion. Recommend HCW training interventions as a priority.
  3. Cultural Nuances: Neglects religious/ethnic drivers of hesitancy (e.g., role of community leaders). Incorporate case studies from diverse cultural contexts.
  4. Technology Access: New technology interventions assume digital literacy. Address accessibility barriers (e.g., rural areas, low-income groups).
  5. Publication Bias: Over 70% of studies are from the past 3 years. Ensure older studies are not disproportionately excluded if methodologically robust.
  6. Visual Aids: Expand tables (e.g., Table 1) to include intervention efficacy metrics (e.g., vaccination rate changes) for clearer comparisons.

Response 2:

  1. COVID-19 Exclusion: Thank you for the comment. We decided not to include studies on the COVID-19 vaccine because of its unique development characteristics and the specific context of its implementation. Comparing a vaccine developed and tested in record time, with a very short track record in the general population, to other vaccines that have been established for much longer introduces significant differences in public perception and confidence. While we understand that distrust of the COVID-19 vaccine may have affected confidence in other vaccines, including it would have biased our results.

Therefore, a study specifically dedicated to the COVID-19 vaccine would be highly beneficial, as it would allow for a more specific understanding of the factors leading to hesitancy. This would also allow for the development of specific interventions tailored to their unique context.

  1. HCW Engagement: We agree with that commentary. Vaccine hesitancy of between HCWs presents a major challenge, given their role as a trusted source of information for the population. Therefore, interventions are crucial to increase their confidence. Accordingly, we have modified the discussion to place greater emphasis on this issue, Please see page 10, lines 375-381:

“In our review, there are only two interventions aimed at addressing hesitancy among HCWs. In addition to the interventions we have mentioned, there are strategies aimed at HCWs to improve their awareness and implementation of best immunization practices. These include team discussions and communication training using a "presumptive communication approach" or motivational inter-viewing [82–84], weekly educational flyers [85], and role-playing or virtual reality simulations [86–88].”.

  1. Cultural Nuances: Thank you for pointing this out. The lack of articles emphasising religious/ethnic factors is an issue that we have taken into account and noted in the limitations of the manuscript, Please see page 10, lines 416-423:

“Other reviews [15,72] have reported the presence of populations whose hesitancy was due to the influence of religious leaders, in which interventions were used in which leaders and health professionals were involved in a coordinated way. However, this reason for hesitancy was not mentioned in any of the articles reviewed and was not analyzed in the review. In addition, the different studies analyzed have been developed in different countries, implying that different cultural contexts may influence the concept of immunization; and there are other factors that may influence hesitancy that were not analyzed in the review, such as educational and socioeconomic level.”.

  1. Technology Access: We appreciate this comment. In the discussion we have talked about the technological advances that have not been described in the articles analysed. However, one factor to take into account is that there are populations that cannot use this type of intervention. It is therefore necessary to develop interventions tailored to the target population, taking into account the cultural, economic and social level, the vaccine and the reasons for reluctance. This idea is described in the conclusion, Please see page 11, lines 442-446:

    “The design of an effective intervention to improve vaccination uptake must include understanding the reasons for hesitancy, developing structured and organized content, tailoring vaccination to the target population and proposing a vaccine, using validated surveys for assessment and employing different types of interventions to achieve greater effectiveness.”.

  2. Publication Bias: Thank you for the comment. We decided to conduct a systematic review of articles published in the last five years because of the context of the global COVID-19 pandemic has been a key factor in amplifying a movement of hesitancy that was already present and now poses a significant public health challenge. In addition, the decision-making process regarding vaccination is profoundly influenced by the social context, including beliefs, perceptions and considerations related to vaccine availability and costs. Therefore, we have found a higher number of articles in the last 3 years, due to the presence of the COVID-19 pandemic and the increased hesitancy following the development of the vaccine.

  3. Visual Aids: Thank you for the comment. We have a summary of the main methods for measuring effectiveness in results, Please see page 6, lines 190-193:

“To assess efficacy, 37% of the studies compared vaccination rates before and after the intervention. The remaining 63% relied on pre- and post-intervention questionnaires, of which only five [29,33,47,58,60] had been validated or peer-reviewed.”.

As well as a table with the main advantages of each type of intervention according to our results to emphasize this point and summarize the results. This change can be found on page 9, Table 2.

Reviewer 3 Report

Comments and Suggestions for Authors

The manuscript "Immunity awareness - Strategies to Improve the Degree of Acceptance of Vaccines: a Systematic Review" by Alejandro Martínez-Serrano et al., reviews the possible strategies to improve the level of vaccine acceptance toward different target populations. I agree with the authors that this manuscript provides valuable insights about how to reduce vaccine hesitancy, particularly when dealing with different target populations.

Here are my major concerns:

1. Authors provide a long summary of the results but lack interpretation and fail to correlate the findings within the context of existing knowledge and future directions. The author should incorporate a discussion section to enhance the impact of their manuscript.

2. What are the advantages of using a multicomponent intervention compared to a single intervention in addressing vaccine hesitancy? Can the author identify any qualitative or quantitative evidence from the selected literature?

Author Response

Point-by-point response to Comments and Suggestions for Authors

Comments 1: The manuscript "Immunity awareness - Strategies to Improve the Degree of Acceptance of Vaccines: a Systematic Review" by Alejandro Martínez-Serrano et al., reviews the possible strategies to improve the level of vaccine acceptance toward different target populations. I agree with the authors that this manuscript provides valuable insights about how to reduce vaccine hesitancy, particularly when dealing with different target populations.

Here are my major concerns:

1. Authors provide a long summary of the results but lack interpretation and fail to correlate the findings within the context of existing knowledge and future directions. The author should incorporate a discussion section to enhance the impact of their manuscript.

Response 1: Thank you for pointing this out. We have faced several circumstances as the reviewer suggested due to the variability of the populations included, type of interventions, etc. Therefore, we have included a paragraph in the discussion section to articulate the direct implications or our results and to enhance the impact of the manuscript. It can be found in pages 10 and 11, lines 401-412:

Consequently, to improve evaluation and facilitate meaningful comparisons of short- and medium-term impacts, it is necessary to consider the domains of the BeSD model and use validated pre-post intervention surveys designed to measure vaccination hesitancy in various populations and vaccines, such as the Vaccination Confidence Index [95] or the Vaccination Attitudes Examination (VAX) [96]. Our study explores the effectiveness of different interventions that improve vaccine acceptance, addressing the causes and different dimensions behind vaccine hesitancy, as an essential prior step for the formulation of public health policies to improve vaccine coverage in a context-sensitive manner [97], which improve trust in healthcare professionals' interventions [98]. Further research is needed to evaluate the long-term objectives of interventions aimed at reducing vaccine hesitancy, considering vaccination rates in order to track trends within the general population.

Comments 2: 2. What are the advantages of using a multicomponent intervention compared to a single intervention in addressing vaccine hesitancy? Can the author identify any qualitative or quantitative evidence from the selected literature?

Response 2: We appreciate the comment. Although direct comparison between the results is not feasible, due to the lack of comparable quantitative data in many of them, the different populations addressed and the variability of interventions.  The WHO Strategic Advisory Group of Experts on Immunization (SAGE) working group report analyzing various interventions highlights that multicomponent approaches tend to be more effective, as stated in the introduction section, Please see page 2, lines 70-73:

“… the WHO Strategic Advisory Group of Experts on Immunization (SAGE) working group (16) developed an analysis which concluded that the most effective interventions were those that use multiple strategies (multicomponent interventions) to enhance vaccine uptake, knowledge, and awareness.”.

Moreover, there is evidence that many interventions are rarely implemented in isolation; rather, they often serve to complement and support others — for example, the use of mass media typically functions as an adjunct to broader strategies. That is what we found in this systematic review, Please see page 9, Table 2.

For example, one of the interventions that introduces several strategies is the Comfort Ask Relax Distract (CARD) system, which is a vaccine delivery framework that integrates evidence-based interventions to reduce stress-related responses and im-prove the vaccination experience. The CARD system has shown good results but in a specific setting (schools, adolescents).

Thus, the integration of different types of strategies in the intervention allows the ideas developed to be reinforced and complemented in various ways, reinforcing the acquisition of knowledge and increasing acceptance.

Round 2

Reviewer 2 Report

Comments and Suggestions for Authors

The authors have addressed all my concerns.

Reviewer 3 Report

Comments and Suggestions for Authors

Thank you for the point-by-point responses. I am ok with the revised version of the manuscript.